# Tailing Optical Pulling Force on a Metal–Dielectric Hybrid Dimer with Electromagnetic Coupling

**DOI:** 10.3390/nano13152254

**Published:** 2023-08-05

**Authors:** Xiao-Ming Zhang, Jin-Jing Yu, Hai-Ping Wu, Xia Zhou, Tian-Yue Zhang, Jian-Ping Liu

**Affiliations:** 1College of Physics Science and Engineering Technology, Yichun University, Yichun 336000, China; 18897958011@163.com (J.-J.Y.); w2707708365@163.com (H.-P.W.); 2College of Literature, Journalism and Communication, Yichun University, Yichun 336000, China; zhouxia_2003@126.com; 3School of Integrated Circuits, Beijing University of Posts and Telecommunications, Beijing 100876, China

**Keywords:** optical pulling force, hybrid nanoparticles, surface plasmon polariton, resonant mode

## Abstract

In this work, we demonstrate that optical pulling forces (OPFs) can be induced by a hybrid dimer consisting of a Si nanoparticle (NP) and a coated nanoparticle with a gain core and Au shell under normal plane wave illumination. Analytical theory reveals that the underlying physical mechanism relies on interactions between the electric dipole (ED) modes excited in the NPs. As compared with the individual NP, it is found that the magnitude of optical force can be enlarged by almost three orders for the Si NP and one order for the coated gain NP in the coupled dimer. In addition, we find that the OPFs exerted on the NPs are heavily dependent on the gain level of the core materials, the incident polarization angle and the sizes of the NPs. More interestingly, we find that the OPF can also be exerted on a trimer system consisting of two identical Si NPs and a coated NP arranged in a line. The related results could be used to propose a versatile platform for manipulating NPs.

## 1. Introduction

Since the demonstration on manipulation with micro-meter size particles reported by Ashkin in the 1970s [1], optical manipulation based on optical force has been well-developed and has opened numerous avenues in fundamental and applied science, e.g., physics, chemistry and biology [2,3,4]. Optical force exerted on small objects, in general, can be decomposed into gradient force and radiation pressure. The former is proportional to the intensity gradient of the incident light and originates from the inhomogeneity of the electromagnetic field, which can manipulate small particles [5]. Radiation pressure, on the other hand, always pushes particles made of a passive medium in the direction of light propagation due to the momentum conservation, thus destabilizing optical trapping. Recently, nonconservative negative optical forces or optical pulling forces (OPFs) have attracted extensive considerable attention due to their unusual behaviors: they can pull particles towards the source of light via a backward scattering force [6,7,8,9,10,11,12,13,14,15]. In fact, it can be rigorously shown that OPF is impossible for a passive nanoparticle (NP) of any shape or composition with a uniform electromagnetic plane wave. However, in some special cases, OPFs can be obtained. For instance, structured beams can provide the means for generating OPF [7,9,10]. In other schemes, the OPF can exert on gain (active) NPs because of an enhancement in the optical linear momentum as the interaction of light occurs with the NPs [11,12,13]. An OPF can also appear when a particle located near the plasmonic interfaces occurs due to the spin–orbit coupling effect [14]. In addition, strong mode coupling arising from exceptional interaction between the resonant modes in NPs has raised much attention, which can also be used to generate OPFs. For example, Lin et al. found that an OPF can act on a plasmonic NP due to the Fano resonance coupling between adjacent plasmon modes [15]. More recently, Xu et al. showed that a small OPF can exert on a low refractive index NP by positioning it close to a plasmonic NP, due to the interplay between the ED and the magnetic dipole [8]. Although intensive studies have been conducted for OPFs on a single dielectric or metal NP [6,7,8,9,10,11,14,15], the intriguing physics and consequences of the light pulling all the interacting NPs in a hybrid dimer, trimer and even polymer, remain elusive and unexplored. In this theoretical paper, we show that induced OPFs can both exert on a Si NP and a core-shell NP when they are placed close to each other, under normal plane wave excitation. We then reveal the underlying physics and trace the origin of this intriguing phenomenon. Finally, we show that the OPF can also be exerted on a hybrid metal–dielectric trimer system.

## 2. Results and Discussion

The schematic of this hybrid dimer is depicted in Figure 1a and the system is assumed to be freestanding in air. The full-wave calculations are performed using the finite-difference time-domain (FDTD) method. The dimer consists of a Si and a core-shell NP. The coated NP is composed of an active core with an inner radius *r* and the Au shell with an outer radius *R*_B_. The refractive index of the core is *n-ik* and the negative *k* denotes the global level of the gain in the materials. We note that the gain effect can be achieved by semiconductor material with external pumping [16]. The radius of the Si NP is *R*_A_ and *g* denotes the surface-to-surface distance. The dimer is illuminated by a plane wave propagating towards the *z* direction, which is polarized along the *x*-axis. The dielectric constant of Au and Si are taken from [17] and [18], respectively.

To quantify the OPFs exerted on the NPs, the calculations for the optical forces can be performed by using the Maxwell stress tensor (MST) method $\overset{↔}{T}$ [13,19]:(1)T↔=12Reεrε0EE∗+μrμ0HH∗−I↔2(εrε0Ε2+μrμ0H2)
where $\varepsilon_{r}$, $\mu_{r}$ are the relative permittivity and permeability. The time-averaged force can be obtained by integrating the MST over the surface *S* of the Si or the coated NP [13,19]:(2)$$\langle F\rangle =\int_{S}\langle \overset{↔}{T}\rangle •nds$$

Figure 1b shows the *z*-component optical forces acting on the dimer, with *R*_A_ = *R*_B_ = 50 nm, *r* = 40 nm, *g* = 60 nm, and the refractive index of the core particle is 1.44-0.345*i*. For the coated NP, it can be seen that the negative value of the *z*-component optical force appears in the whole wavelength band (see black line), indicating that an OPF can be exerted on the coated NP. Interestingly, for the Si NP, an OPF can also be achieved in the range from 618 nm to 621.4 nm. The high-refractive-index NPs, e.g., Si and Ge, are generally difficult to trap due to the mismatch between the NPs and their surrounding medium [15]; however, the OPF acting on the Si NP can be easily realized in our system due to strong electromagnetic coupling, which can be characterized by the near-field intensity distribution in the gap region. We can clearly observe that the near-field intensity in the gap at the wavelength position in the OPF peak (see inset in Figure 1b) is very large, and can induce a strong OPF. We note that the strong OPFs on the NPs mainly come from the resonant interplay between the gain medium and the metallic nanoshell. Furthermore, it is also clearly observed that the OPF peaks for the Si and coated NPs appear at the same wavelength position (about 621 nm). To better understand the physical mechanism of the OPFs, we study the optical response of the isolated NP, as shown in Figure 2a,b. For the individual coated NP, an OPF can be obtained (see black solid line in Figure 2a), due to the recoil generated by the extra momentum in the forward direction and the maximal OPF can be enlarged about two orders larger than that of a single homogeneous gain NP, as has been corroborated theoretically [11,20]. However, the magnitude of the OPF acting on the individual coated NP can only reach about 3.4 pN at the position of the gain-assist plasmon ED mode, which is weakened by about one order compared with that in the coupled case. The case of the Si NP is shown in Figure 2b, where we can clearly see that the OPF is absent (see black line) due to the momentum conservation theory, meanwhile, the magnitude of the OPF is weakened by about three orders compared with that in the coupled dimer. Furthermore, we can clearly observe that the scattering spectra of the Si and coated NPs are both dominated by the contributions of the dipole modes (see red solid lines in Figure 2a,b). It is worth noting that the OPFs can also be extended to other dielectric NPs in the coupled dimer. Furthermore, the OPFs on the NPs appear in a very narrow wavelength range, which may impose limitations for many practical applications, and to avoid this, we can use the silver shell with the same size instead of the Au shell to broaden the OPFs band efficiently (the related results are not shown here).

The results shown above are numerical simulations, but in order to illustrate the physical origin of the behaviors shown in Figure 2, from now on let us work on the OPF analytically. For small NPs, we can employ the dipole approximation and their optical properties can be described by the induced dipoles on NPs [21]:(3)$$p_{A,B}=\varepsilon_{0}\alpha_{A,B}E_{0},m_{A,B}=\chi_{A,B}H_{0}$$
where **P**_A_ (**P**_B_) and **m**_A_ (**m**_B_) are the induced ED and MD moments of the Si NP and coated NP, respectively. **E**_0_ and **H**_0_ are the external electric and magnetic fields. $\alpha_{A,B}$ and $\chi_{A,B}$ are the electric and magnetic polarizabilities, which can be calculated by the Mie coefficients *a*_1_ and *b*_1_ [21]:(4)$$\alpha_{A,B}=6\pi i\frac{a_{1}}{k_{0}^{3}},\chi_{A,B}=6\pi i\frac{b_{1}}{k_{0}^{3}}$$
where *k*_0_ is the wavevector. To investigate the optical response of an interaction coated NP and Si NP, the coupled electric and magnetic dipole approximation (CEMDA) method is applied. The equations can be written as [8,21,22]:(5)$$p_{A}=\alpha_{A}\{\varepsilon_{0}E_{0}+ap_{B}+b(n\cdot p_{B})n+\frac{d}{c}(n\times m_{B})\}$$
(6)$$p_{B}=\alpha_{B}\{\varepsilon_{0}E_{0}+ap_{A}+b(n\cdot p_{A})n+\frac{d}{c}(n\times m_{A})\}$$
(7)$$m_{A}=\chi_{A}\{H_{0}+am_{B}+b(n\cdot m_{B})n-dc(n\times p_{B})\}$$
(8)$$m_{B}=\chi_{B}\{H_{0}+am_{A}+b(n\cdot m_{A})n-dc(n\times p_{A})\}$$
where *c* is the speed of light, **n** is a unit vector and points from the Si to the coated NP, and the coefficients *a*, *b* and *d* can be written as:(9a)$$a=\frac{e^{ikD}}{4\pi D}(k_{0}^{2}-\frac{1}{D^{2}}+\frac{ik_{0}}{D})$$
(9b)$$b=\frac{e^{ikD}}{4\pi D}(-k_{0}^{2}+\frac{3}{D^{2}}-\frac{3ik_{0}}{D})$$
(9c)$$d=\frac{e^{ikD}}{4\pi D}(k_{0}^{2}+\frac{ik_{0}}{D})$$
where *D* is the distance between the NPs. It is pointed out that these solutions can provide us with useful insights and can provide information about the optical responses of the hybrid under different conditions. In our system, the dimer is aligned at the *x*-axis (**n** = (1, 0, 0)) and the plane wave is propagated along the *z*-axis (**k** = (0, 0, *k*)), with the electric field polarized along the *y*-axis (**E**_0_ = (0, *E*_y_, 0)). In this configuration, one may note that all the components, except *p*_Ax_, *p*_Bx_, *m*_Ay_, *m*_By_, *p*_Az_, and *p*_Bz_, vanish. Once the induced ED and magnetic dipole (MD) moments of the NPs are obtained, the optical force exerted on each particle in the coupled system reads as [8,22]: (10)F=12Re{(∇E0∗)⋅p+(∇H0∗)⋅m−k046πε0c(p×m∗)}

In Equation (10), **F** can be decomposed into three components, i.e., **F***_e_*, **F***_m_* and **F***_em_*. Physically, **F***_e_* and **F***_m_* arise from the dipole moments **p** and **m**, respectively. The component **F***_em_* arises from the interaction between **p** and **m**. Figure 2c,d shows the results for *F*_z_ calculated by the CEMDA (green dashed lines) acting on the coated and Si NPs, respectively. They show excellent agreement with the results obtained by the FDTD calculations (black solid line). Additionally, we see that *F_e_* indeed plays a dominant role in *F*_z_, while *F_m_* and *F_em_* can be negligible due to their small magnitudes for both NPs. This can be easily explained with the help of multipole expansion, as shown in Figure 2a,b. For both NPs, the MDs can be neglected due to their very small contributions to the scattering compared with that of EDs. In such case, *F*_z,A_ and *F*_z,B_ are mainly determined by the imaginary part of *p*_x,A_ and *p*_x,B_, respectively:(11)Fz(A,B)=12k0Im[px(A,B)]E0

It implies that *F*_z_ should be proportional to the imaginary of *p*_x_ once the wavelength is fixed. In Figure 3a, we show the dependence of *F*_z_ on the *g*, exerting upon the Si NP (red solid line) and coated NP (black solid line), respectively. Figure 3b shows the corresponding imaginary part of *p*_x_ excited in the coated NP (black dashed line) and Si NP (red dashed line). The incident wavelength is fixed at 621 nm. As anticipated in Equation (11), the line shapes for *F*_z_ are similar to the corresponding *p*_x_ for both NPs. Furthermore, it can be observed in Figure 3a that an optical pushing force can act on the coated NP when the *g* is less than 57 nm, and a remarked OPF appears when the *g* increases further; in addition, the maximum OPF emerges at around *g* = 60 nm. As the *g* increases, it decays when approaching about 3.4 pN, which is the same as the result for the individual coated NP (see black sold line in Figure 2a). This is because when the gap size *g* becomes very large, the optical response of the resulting *p*_B_ in the coupled dimer can be considered the same as that of the individual coated NP due to a, b → 0. For the Si NP, the OPF appears in the range from 30 nm to 120 nm, where the maximum OPF occurs at around *g* = 59 nm, and decays approaching a small optical pulling force when *g* is larger than 120 nm.

Furthermore, the induced ED can also be strongly influenced by the gain level [11,12]. In Figure 4a,b, we show the results for the OPFs as a function of the wavelength and gain parameter *k* exerted on the Si NP and core-shell NP, respectively. All the other parameters are identical with that of Figure 1b. The white and colored regions in Figure 4 represent the optical pushing force and OPF, respectively. For the coated NP, as shown in Figure 4b, an optical pushing force is exerted on the NP when *k* is less than 0.345 and the OPF appears beyond that. When *k* increases further, the amplitude of the OPF decreases and the maximum OPF occurs at about *l* = 621 nm for *k* = 0.345. This is because when the gain parameter *k* is increased after the singular point, the magnitude of the *p*_B_ degrades due to excess gain [12,13]. As shown in Figure 4a, we can clearly observe that the optical pushing force acts on the Si NP when *k* is less than 0.24 in the whole range of wavelengths. Beyond that, the OPF appears and the OPF wavelength band becomes wide as *k* increases. As *k* increases to about 0.4, the OPF wavelength band gradually becomes narrow as *k* increases further. We can also observe that the maximum OPF acting on the Si NP also appears at *λ* = 621 nm for *k* = 0.345, this arises from the fact that the induced *p*_A_ of the Si NP in Equation (5) is totally determined by the *p*_B_ coated NP when the coefficients *a* and *b* in Equation (6) are non-negligible.

Next, we investigate the other geometry parameters that can affect the OPF exerted on the core-shell NP and Si NP in the coupled dimer, including the incident polarization angle (*θ*), and the sizes of the NPs (*R*_A_, *R*_B_ and *r*). In Figure 5a,b, we show the results for the OPFs with different *θ* exerted on the core-shell NP and Si NP, respectively. In fact, the induced ED moments in both the Si and core-shell NPs could be changed when the polarization angle *θ* is varied [8] therefore, the near-field coupling between the two NPs will be varied too. From Figure 5a,b, we notice that both the Si NP and core-shell NP are subject to OPFs along the propagation direction when the polarization angle *θ* is 0°, 30° and 60°. Furthermore, the magnitudes of the OPFs gradually decrease as the polarization angle *θ* increases and the wavelength positions for the maximum OPF is fixed at about 621 nm. When *θ* = 90°, it is observed that the Si NP is subject to an optical pushing force (see green dashed line in Figure 5b), while the core-shell NP is subject to a weak OPF, this is because the electromagnetic interaction between the two induced ED moments is minimal in such a case. In order to study the effects of the size of the core-shell NP on the OPFs, the *z* component of the optical force exerted on the NPs with different *R*_B_ and *r* has been calculated, as shown in Figure 6a–d. All the other parameters are identical with that of Figure 1b. We can clearly see the maximal OPFs appear at *R*_B_ = 50 nm (see Figure 6a,b) and *r* = 40 nm (see Figure 6c,d) at about 621 nm, while a significant weak optical force occurs for the other sizes. This is because the strong ED in the core-shell NP with a gain core and a plasmonic shell is excited at certain geometrical parameters (singular point) and the small changes in sizes will lead to dramatic variation in the optical force [12]. Hence, one could optimize the core-shell particle’s structure parameters to realize optical manipulation. In Figure 6e,f, we show the OPFs exerted on the core-shell NP and Si NP with different radius for the Si NP (*R*_A_), respectively. For the core-shell NP, the OPF appears when the *R*_A_ is varied from 30 nm to 50 nm and the magnitude gradually increases when the *R*_A_ increases in the interested wavelength range; however, the optical force is positive (i.e., the pulling force) when the *R*_A_ is larger than 50 nm. For the Si NP, we can see that the OPF can be exerted on this NP with different *R*_A_ and the maximal OPF occurs at *R*_A_ = 50 nm.

The electromagnetic coupling induced OPF is not solely limited to a dimer system, but more generally, it can occur in the case of multipole closely spaced NPs in a hybrid system, which can offer an alternative strategy to manipulate multipole NPs. Figure 7a,b shows the optical forces that act on the coated NP (denoted as B) and Si NPs (denoted as A and C) in a trimer system, respectively. The hybrid metal–dielectric system consists of two identical Si NPs, which are symmetrically placed on each side of a coated NP with a distance of *l* (50 nm, 60 nm and 70 nm). It is noted that the responses of the two Si NPs are the same due to the symmetry of the system. It can be observed that notable OPFs can act on all the three NPs when the incident wavelength ranges from 600 nm to 623 nm, originating from the same physical mechanism mention above, as can be established from the corresponding induced ED moment spectra displayed in Figure 7c,d. It is also observed that the magnitude of the OPF is gradually decreased as *l* increases, as shown in Figure 7, this is because the mutual interaction through the light exchange weakens as the coefficients *a* and *b* decrease [8]. In addition, we see that the OPF peak positions are gradually blue shifted as l increases due to the mode hybridization [12]. Finally, we investigate the system when the dimer is illuminated by a focus beam. Figure 8a shows schematics of the NPs illuminated by a Gaussian beam propagated along the *z*-axis, which is polarized along the dimer axis (*x*-axis), with the beam width set as 1 mm, and the intensity of the Gaussian beam is 1 mW, and *h* is the distance between the beam center and the dimer axis. In Figure 8b, we show the force as a function of the wavelength with different *h*, and we can see that the OPFs exerted on the NPs can also be obtained in the range from 618 nm to 621.3 nm by the Gaussian beam. Furthermore, the peak wavelengths of the OPF are almost fixed at about 621 nm by the variations of *h*. The related results could be used to propose a versatile platform for manipulating NPs.

## 3. Conclusions

In conclusion, we have demonstrated that OPFs can be induced by a hybrid dimer consisting of Si and coated NPs with a gain core and Au shell. Analytical theory reveals that the underlying physical mechanism of the OPFs arises from interactions between the ED modes excited in the NPs. As compared with a single NP, it is found that the magnitude of the optical force can be enlarged by almost three orders for the Si NP and one order for the coated gain NP in the hybrid dimer. We also show that an OPF can be exerted in a metal–dielectric trimer system. Finally, we show that an OPF can be exerted on the Si and coated NPs when the system is illuminated by a laser Gaussian beam. The interesting results add a novel degree of freedom to optical manipulation.

## Figures and Tables

**Figure 1 nanomaterials-13-02254-f001:**
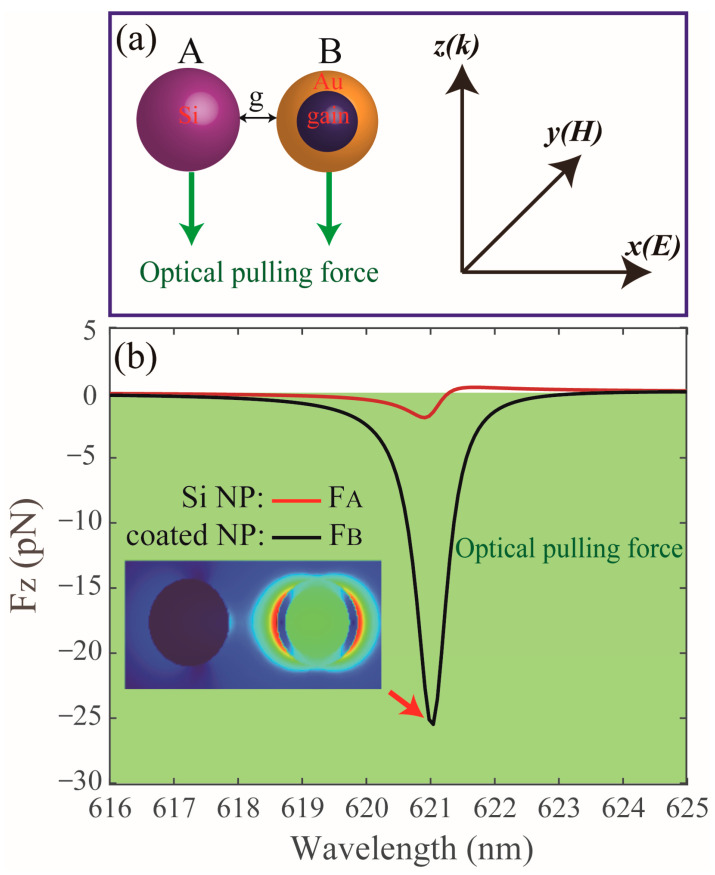
(**a**) Schematic of OPF induced on a dimer consisting of a Si and a core-shell NP. The coated NP is composed of an active core with an inner radius *r* and the refractive index is *n*-*ik*, and the Au shell with an outer radius *R*_B_. The radius of the Si NP is *R*_A_ and *g* is the surface-to-surface distance. The intensity of the incident light is 1 mW/mm^2^. (**b**) The *z*-component optical forces act on the dimer with *R*_A_ = *R*_B_ = 50 nm, *r* = 40 nm, *g* = 60 nm, and the refractive index of the core particle is 1.44-0.345*i*. The inset in (**b**) shows the near-field intensity distribution at the wavelength position in the OPF peak (*λ* = 621 nm).

**Figure 2 nanomaterials-13-02254-f002:**
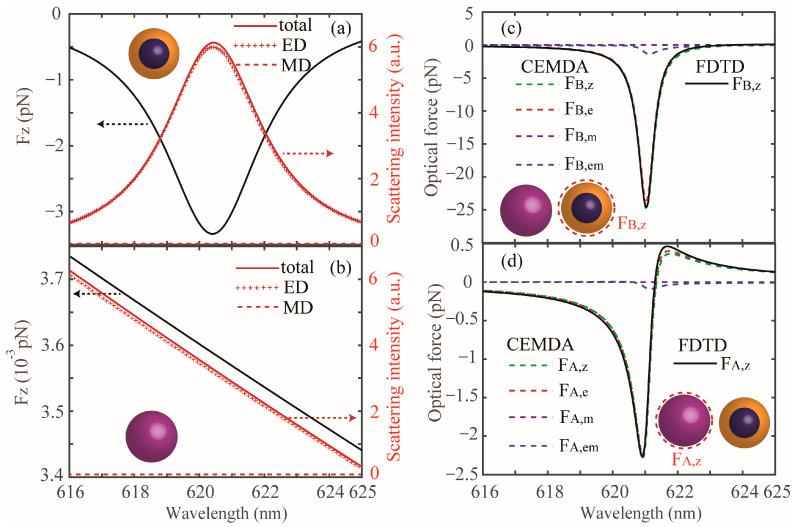
(**a**,**b**) *z*-component optical forces (black lines) and scattering intensity (red solid lines) of the individual NP. The scattering intensity of the electric dipole modes (red dotted lines) and magnetic dipoles (brown dashed lines) are also presented. (**c**,**d**) Optical force (green dashed lines) and three components *F*_e_ (red dashed lines), *F*_m_ (brown dashed lines), *F*_em_ (blue dashed lines) acting on the NPs in the dimer using the CEMDA method. The black solid lines represent the *z*-component optical forces by the FDTD method. The geometrical parameters are identical with that of Figure 1b.

**Figure 3 nanomaterials-13-02254-f003:**
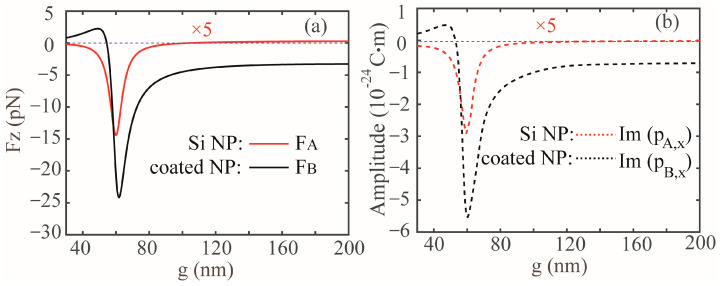
(**a**) Optical force *F*_z_ acting on a coated NP (black solid line) and Si NP (red solid line) versus the surface-to-surface distance *g*. (**b**) Imaginary part of *p*_x_ excited in the coated NP (black dashed line) and Si NP (red dashed line) versus the surface-to-surface distance *g*. All other parameters are identical with that of Figure 1b, and the incident wavelength is set at 621 nm. The blue dashed lines denote the zero OPFs.

**Figure 4 nanomaterials-13-02254-f004:**
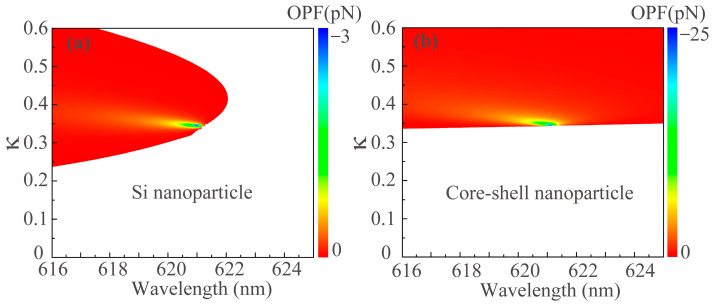
OPF diagrams with respect to the illumination wavelength and gain parameter *k* on (**a**) Si NP and (**b**) core-shell NP in the coupled dimer, respectively. All the other parameters are identical with that of Figure 1b. The white regions indicate the parameters for the optical pushing force, while the colored regions represent the parameters for the OPF.

**Figure 5 nanomaterials-13-02254-f005:**
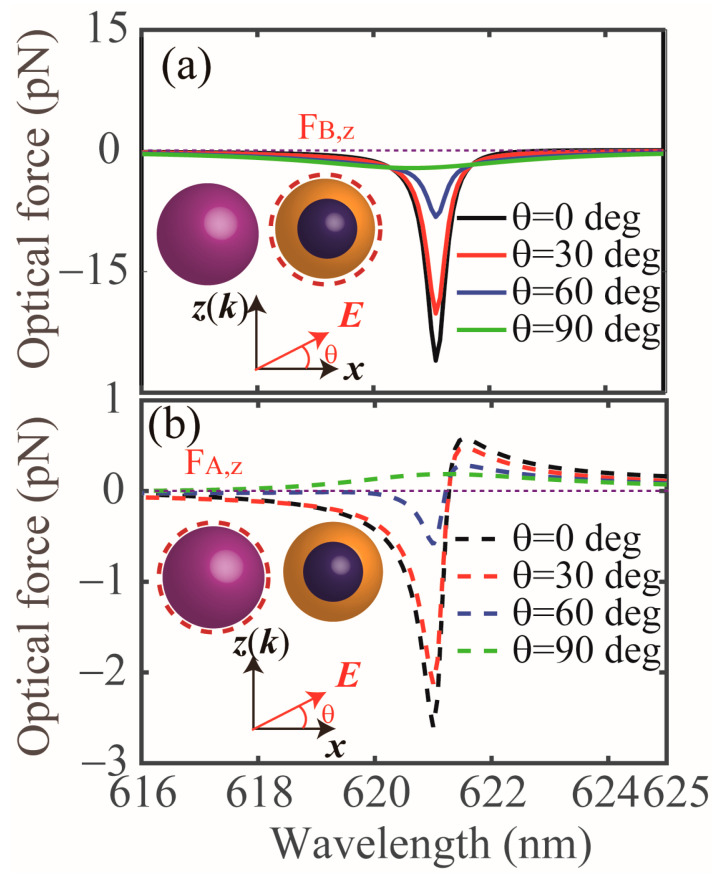
Optical forces on (**a**) the core-shell NP and (**b**) the Si NP in a coupled dimer with different incident polarization angles (*θ*), respectively. All the other parameters are identical with that of Figure 1b.

**Figure 6 nanomaterials-13-02254-f006:**
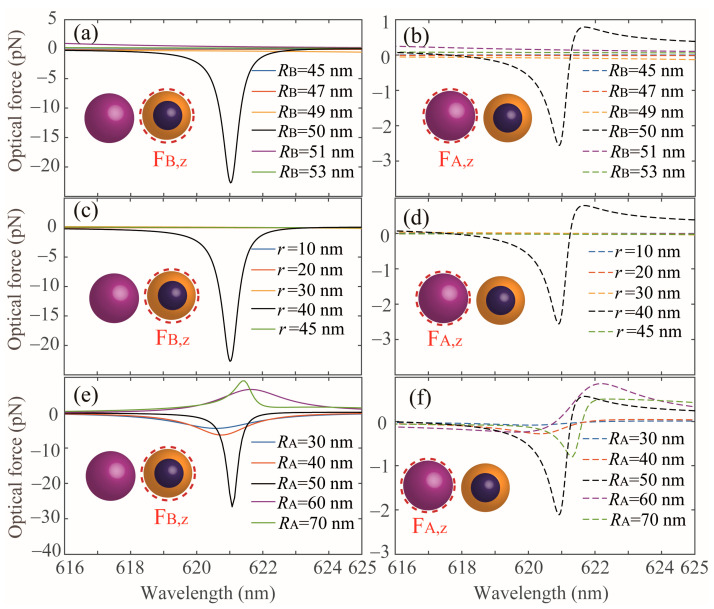
Optical forces on (**a**,**c**,**e**) the core-shell NP and (**b**,**d**,**f**) the Si NP in a coupled dimer with different sizes (*R*_B_, *r* and *R*_A_), respectively. All the other parameters are identical with that of Figure 1b.

**Figure 7 nanomaterials-13-02254-f007:**
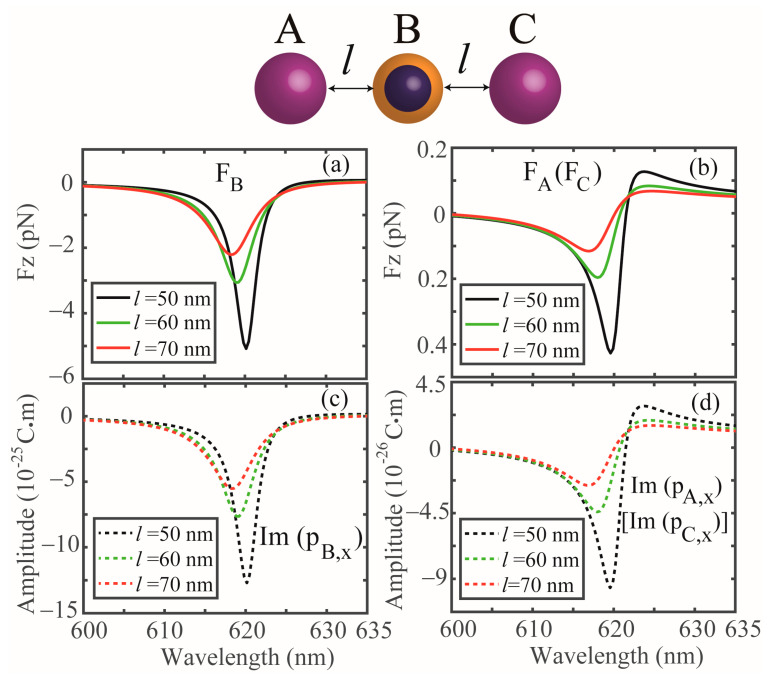
(**a**,**b**) Optical forces in a hybrid trimer system with different distances *l* (*l* = 50 nm, *l* = 60 nm and *l* = 70 nm). (**c**,**d**) Imaginary part of the induced ED moments for the coated NP and Si NPs, respectively. All the other parameters are identical with that of Figure 1b.

**Figure 8 nanomaterials-13-02254-f008:**
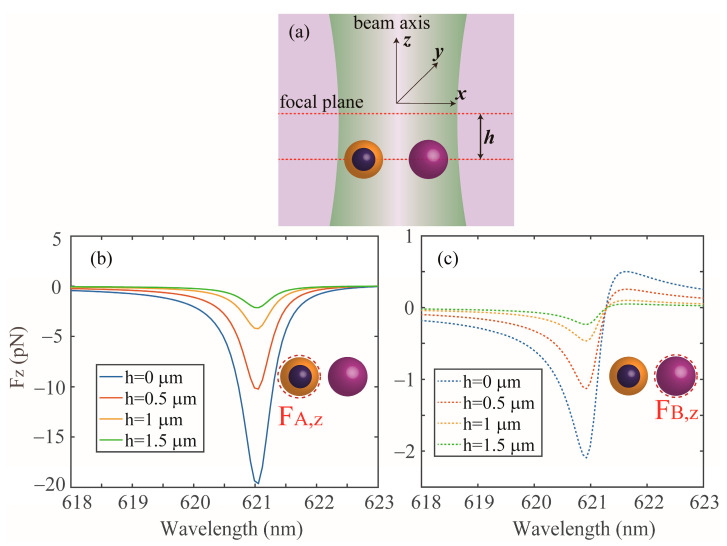
(**a**) Optical force exerted on the coated and Si NPs illuminated by a Gaussian beam. The beam width is 1 mm and the intensity is 1 mW. The geometrical parameters are the same as Figure 1b. (**b**,**c**) *F_z_* exerted on the coated and Si NPs with different *h*, respectively.

## Data Availability

The data presented in this study are available on request from the corresponding author.

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
