# Peer review of "Tailing Optical Pulling Force on a Metal–Dielectric Hybrid Dimer with Electromagnetic Coupling"

_nanomaterials, 2023, doi:10.3390/nano13152254_

Round 1

Reviewer 1 Report

Paper by Zhang, X.-M.; Yu J. J.; Wu, H.-P.; Zhou X..; Zhang, T, Y, Liu J. P. entitled “Tailing optical pulling force on a metal-dielectric hybrid dimer with electromagnetic coupling” is devoted to optical pulling force in a specific dimer and trimer. The theoretically analyzed system is essentially virtual, but it is worth consideration. The results are remarkable among other research in the field. The paper can be published after minor revision.

1. First statement of the paper is obviously incorrect: radiation pressure was already recognized by Kepler  in 1619.

2. line 125: Unclear sentence  “a and c are electric and magnetic polarizabilities”

3. line 153-154: misprints  

Authors/editors should check other misprints over the manuscript.

Paper by Zhang, X.-M.; Yu J. J.; Wu, H.-P.; Zhou X..; Zhang, T, Y, Liu J. P. entitled “Tailing optical pulling force on a metal-dielectric hybrid dimer with electromagnetic coupling” is devoted to optical pulling force in a specific dimer and trimer. The theoretically analyzed system is essentially virtual, but it is worth consideration. The results are remarkable among other research in the field. The paper can be published after minor revision.

1. First statement of the paper is obviously incorrect: radiation pressure was already recognized by Kepler  in 1619.

2. line 125: Unclear sentence  “a and c are electric and magnetic polarizabilities”

3. line 153-154: misprints  

Authors/editors should check other misprints over the manuscript.

Reviewer 2 Report

General comments:

The manuscript by X.-M. Zhang et al. presents the optical pulling force (OPF) can be induced in a hybrid dimer system consisting of a silicon (Si) nanoparticle (NP) and a gain dielectric core coated with a plasmonic material under a plane wave illumination at normal incidence. The manuscript is supported by a nice introduction, and investigations to support their assertions and conclusions. The manuscript could appeal to the related audiences. To be published the following issues be addressed.

Suggestions for revision:

1) It has been investigated that the OPF can be observed in a graded plasmonic core-shell nanoparticle (NP) consisting of a gain dielectric core [1] and graded plasmonic shell or a hybrid dimer system consisting of a gold (Au) NP and a silica (SiO2) NP [2]. So, it would be informative if the authors can provide whether the gain dielectric or the plasmonic NP in the hybrid dimer system predominontly contributes to the OPF.

[1] Physics 3(4), 955-967 (2021).

[2] Optics Letters 43(20), 4961-4964 (2018).

2) What is the reason for choosing Si as the dielectric and Au as the plasmonic shell?

The dependence of the OPF on the refractive index of Si (at least low and high index of refraction) should be investigated.

3) The dependence of the OPF on the geometrical parameters (e.g., RA, r, RB, etc) of the metal-dielectric hybrid dimer system should be studied.

4) Due to the geometry of the hybrid dimer structure, it would show polarization dependent properties. The dependence of the OPF on the polarization state of incident light should be explored.

5) The OPF appears in a very narrow wavelength range, which may impose limitations in many applications. It would be informative if the authors can provide the discussion on how the wavelength range can be broadened.

6) On lines 103-104, what is the strong electromagnetic coupling? It would be instructive if the authors can provide field (probably near-field) intensity distributions at the wavelength of 621 nm.

- English editing is required throughout the manuscript because there are many missing articles.

Round 2

Reviewer 2 Report

All the issues have been well addressed, and the manuscript is properly organized and well written. Thus, I recommend the manuscript to be accepted for publication now.